# Value of Repeat CT Brain in Mild Traumatic Brain Injury Patients with High Risk of Intracerebral Hemorrhage Progression

**DOI:** 10.3390/ijerph192114311

**Published:** 2022-11-02

**Authors:** Farizal Fadzil, Amy Khor Cheng Mei, Azudin Mohd Khairy, Ramesh Kumar, Anis Nabillah Mohd Azli

**Affiliations:** 1Department of Surgery, Faculty of Medicine, Universiti Kebangsaan Malaysia, Cheras, Kuala Lumpur 56000, Malaysia; 2Department of Surgery, Hospital Tengku Ampuan Rahimah, Klang 41200, Selangor, Malaysia

**Keywords:** mild traumatic head injury, scheduled repeat CT brain, progression of bleed

## Abstract

Patients with mild traumatic brain injury (MTBI) with intracerebral hemorrhage (ICH), particularly those at higher risk of having ICH progression, are typically prescribed a second head Computer Tomography (CT) scan to monitor the disease development. This study aimed to evaluate the role of a repeat head CT in MTBI patients at a higher risk of ICH progression by comparing the intervention rate between patients with and without ICH progression. Methods: 192 patients with MTBI and ICH were treated between November 2019 to December 2020 at a single level II trauma center. The Glasgow Coma Scale (GCS) was used to classify MTBI, and initial head CT was performed according to the Canadian CT head rule. Patients with a higher risk of ICH progression, including the elderly (≥65 years old), patients on antiplatelets or anticoagulants, or patients with an initial head CT that revealed EDH, contusional bleeding, or SDH > 5 mm, and multiple ICH underwent a repeat head CT within 12 to 24 h later. Data regarding types of intervention, length of stay in the hospital, and outcome were collected. The risk of further neurological deterioration and readmission rates were compared between these two groups. All patients were followed up in the clinic after one month or contacted via phone if they did not return. Results: 189 patients underwent scheduled repeated head CT, 18% had radiological intracranial bleed progression, and 82% had no changes. There were no statistically significant differences in terms of intervention rate, risk of neurological deterioration in the future, or readmission between them. Conclusion: Repeat head CT in mild TBI patients with no neurological deterioration is not recommended, even in patients with a higher risk of ICH progression.

## 1. Introduction

Traumatic brain injury (TBI) is a frequent occurrence. In 2018, a paper published in the Journal of Neurosurgery referred to TBI as a “silent epidemic” [1], and their model estimated that 64 to 74 million new TBIs were reported annually worldwide [1]. The study found that the leading risk factor for TBI was road traffic accidents, most prevalent in Southeast Asian and African nations [1]. While in Malaysia, TBI accounts for up to 7.86% of hospitalizations in Malaysian Ministry of Health (MOH) hospitals [2].

Mild traumatic brain injury (MTBI) accounts for the vast majority (80%) of TBIs. Head computed tomography (CT) scan is the preferred diagnostic tool in the management of patients with TBI as it allows for rapid and accurate detection of intracerebral hemorrhage (ICH) [2] and is frequently included in the general multi-trauma work-up [2,3]. However, the need for and the time interval between repeated head CT, the duration of observation, subsequent follow-up, and management vary greatly based on the neurosurgeon’s practice in the region [2,3,4].

Additionally, for patients with neurological deterioration, defined as a decrease in GCS ≥ 2, change in pupillary size, or the onset of symptoms attributable to increased intracranial pressure (ICP) [2,5,6], a repeat head CT is warranted to re-evaluate the magnitude and evolution of the ICH. Yet, for MTBI patients with no neurological deterioration, repeat head CT has a limited ability to predict the need for intervention [5,6,7,8,9,10,11,12,13]. Furthermore, Sifri et al. and Brown et al. found no significant difference in the intervention rate between patients whose CT brain was repeated due to neurological deterioration versus those without [5,7,8]. GCS assessment is the most accurate predictor of the need for intervention [2,8].

Generally, repeat head CT is performed in many centers for a subset of MTBI patients with a higher risk of ICH progression. Cepeda et al. discovered an increased risk of ICH progression in older patients and those with initial ICH of less than 5 mL and subdural hemorrhage (SDH) [14]. Moreover, the MOH guideline recommended inpatient monitoring for elderly patients (≥65 years old) or those receiving antiplatelet or anticoagulant therapy in conjunction with MTBI [2]. Likewise, Menditto et al., Swap et al., and Miller et al. discovered that elderly patients taking anticoagulants had delayed ICH progression [15,16,17]. Swap et al., Kaen et al., and Ohm et al. found an increased risk of delayed ICH in elderly patients taking antiplatelet therapy despite having a negative initial head CT [18,19,20].

Concurrently, multiple studies have identified an increased risk of ICH progression in patients with extradural hemorrhage (EDH), contusional bleeding, or mixed lesions [7,10,21,22,23,24]. Some have proposed that patients with traumatic subarachnoid hemorrhage (SAH) should undergo repeat head CT because this lesion tends to progress and worsen [24,25,26,27].

This study aimed to evaluate the role of a repeat head CT in MTBI patients at a higher risk of ICH progression by comparing the intervention rate between patients with and without ICH progression. Based on the aforementioned studies and recommendations, we decided to perform repeat head CT within 12 to 24 h for MTBI patients with a higher risk of ICH progression; those who were elderly (≥65 years old), on long-term antiplatelets or anticoagulants, or who have worrisome ICH (EDH/SAH/SDH > 5 mm) or multiple ICH. We hypothesized that a repeat head CT in patients with normal neurological examination following MTBI is unlikely to prompt neurosurgical intervention or a change in management, even in high-risk patients.

## 2. Materials and Methods

The National Medical Research and Ethics Committee approved this prospective cohort study and the National University of Malaysia Research Ethics Committee.

In this study, consecutive sampling was used for patients with mild TBI (GCS 13–15), and ICH admitted to the level II trauma center, Hospital Tengku Ampuan Rahimah (HTAR). The data collection took 11 months, from November 2019 to December 2020. A sample of 192 patients was selected according to the Pike and Smith formula.

MTBI is defined as a GCS score between 13 and 15 accompanied by loss of consciousness, post-traumatic retrograde amnesia, headache, vomiting, or dizziness. The identified patients received neurosurgical consultations.

Exclusion criteria included: (a) patients who died prior to the CT scan, (b) patients who were admitted to surgery based on the findings of the first CT scan, e.g., EDH greater than or equal to 30 mL, SDH greater than or equal to 1.5 cm thickness, or the presence of cerebral edema and midline shift, and (c) patients who were discharged or died after the first CT scan.

Accordingly, the primary outcome was to compare the intervention rate of mild TBI (MTBI) patients with radiological progression of ICH in repeated head CT and those without. The secondary outcome includes the readmission rate of MTBI patients and the risk of further neurological deterioration in patients with ICH progression than those without.

### 2.1. Study Protocol

Head CT was performed for patients who presented with GCS 13 to 15 according to the Canadian CT head rule [2]. Patients with a higher risk of ICH progression, including the elderly (≥65 years old), patients on antiplatelet or anticoagulants, those whose initial head CT revealed EDH or contusional bleed or SDH > 5 mm, and those with multiple ICH underwent repeat head CT within 12 to 24 h. The collected variables were age, gender, mechanism of injury, and GCS score on admission for all patients. The type, site, and numbers of ICH were also recorded.

Medical officers and nurses competently trained to evaluate GCS in MTBI patients measured GCS and pupillary size every hour. Those who displayed neurological deterioration prior to their scheduled repeat head CT underwent an urgent head CT. Neurological deterioration was defined as a decrease in GCS score of at least 2 points, a change in pupillary size (>2 mm or unequal pupil size), or the sudden onset of symptoms attributable to head injuries, such as headache, vomiting, dizziness, and visual disturbance.

The radiologist analyzed all repeat head CT and the radiological changes were recorded. ICH progression was defined as the expansion of existing lesions or the onset of a new bleed. The patients were observed for 48 to 72 h before being discharged home if there was no neurological deterioration. Types of intervention, whether medical or surgical, the length of hospitalization and ICU stay, and mortality was documented. All patients were seen in the clinic again one month after being discharged or contacted by phone if they did not return. The study flow is shown in diagrammatic flow chart in Figure 1.

### 2.2. Statistical Analysis

Statistical analysis was performed using SPSS software, version 21.0 (Armonk, NY, USA: IBM Corp.). After obtaining data, including intervention, neurological deterioration during admission, and the patients’ readmission rate, a descriptive analysis was performed to provide demographic information and baseline ICH in the initial head CT. We examined the association of ICH progression with intervention rate, neurological deterioration during admission, and readmission rate using the Chi-square test with Yate’s correction. Statistical significance is defined as *p* < 0.005 for all comparisons.

### 2.3. Ethical Consideration

This was a prospective cohort study conducted from November 2019 to December 2020. The study protocol was approved by the National Medical Research and Ethics Committee (Approval Code: NMRR-19-2942-49474) and the National University of Malaysia Research Ethics Committee. (Approval code: FF-2020-075).

## 3. Results

During the 11-month study, 192 patients sustained MTBI and had a positive initial head CT. Two patients necessitated immediate neurosurgical intervention due to a drop in GCS prior to the repeat head CT. Both underwent emergency decompressive craniectomy and were excluded from the study. One patient died before the repeat head CT and was excluded. The remaining 189 patients who underwent a repeat head CT has an age range from 16 to 101 years old, with a mean age of 45.4. They initially presented with GCS between 13 to 15. After the repeat head CT, 34 patients (18%) were identified to have ICH progression.

In this study population, repeat head CT was performed within 12 to 24 h for patients with a higher risk of ICH progression. This includes the elderly (≥65 years old), patients on antiplatelets or anticoagulants, patients whose initial head CT revealed EDH or contusional bleed or SDH ≥ 5 mm, and those with multiple ICH. 29.6% (*n* = 56) of the population consists of patients older than 65 years of age. Similarly, 13.2% (*n* = 25) of the patients were taking antiplatelet medication.

The demographic characteristics of the study population with ICH progression in repeat head CT and those without are shown in Table 1. We ran Levene’s test to determine population variances in each group. Both groups have comparable population variances (F = 0.25, *p* = 0.62). The most common mechanism of injury was motor vehicle accidents (*n* = 132), followed by falls (*n* = 430, and other injuries (*n* = 14).

Multiple ICH was the most common type of ICH observed on first head CT (43.4%), followed by SDH (31.2%), contusional bleed (15.8%), SAH (5.8%), and EDH (3.7%). Table 2 shows ICH types in relation to ICH progression and those without.

Table 3 demonstrates no statistically significant relationship between the progression of ICH on scheduled repeat head CT and intervention (2.9% versus 0%, *p* = 0.18). Even if there were changes in the repeat head CT, none of the patients without neurological deterioration required surgical or medical intervention.

The timing of the first head CT was recorded as within 6 h and after 6 h. Table 4 showed no correlation between the timing of the first CT scan and the progression of bleeding in repeated head CT (20.5% versus 18.5%, *p* = 0.73). The two patients who sustained a drop in GCS > 2 prior to the repeat head CT had their first head CT performed within 6 h of trauma. The GCS of one patient decreased from 14 to 10 as repeated head CT revealed expansion of EDH. Another had GCS dropped to 9 due to a larger occipital contusional bleed and hydrocephalus. Both underwent urgent decompressive craniectomy. The ages of the two patients were 18 and 21 years, respectively.

Meanwhile, ten patients (5.3%) exhibited neurological deterioration during hospitalization following the scheduled repeat head CT. This deterioration of neurological status was characterized by persistent or increasing confusion, visual disturbances, disorientation, or sleepiness, as well as an increase in the severity of presenting symptoms. On admission, patients with ICH progression did not have an increased risk of neurological deterioration during admission (5.9% versus 5.2%, *p* = 1.0) (Table 5). Thus, there was no correlation between the progression of bleeding on the scheduled repeat head CT and future neurological deterioration.

Eight patients were readmitted after discharge, primarily due to persistent symptoms such as nausea, vomiting, constant headache, and dizziness. There was no significant correlation between ICH progression on repeated head CT and the patient’s readmission rate. Patients with ICH progression did not have a higher readmission rate than those without (5.9% versus 3.9%, *p* = 0.97) (Table 6).

## 4. Discussion

Multiple prospective studies have conclusively demonstrated the value of performing head CT in patients with MTBI, which is now considered the standard of care [3,6,11,16,28]. However, a systematic review and meta-analysis by Reljic et al. found the utilization of repeat head CT in MTBI patients without neurological deterioration remains unproven [3,5,6,8,9,10,11,12].

Rosen et al., Almenawar et al., Wong et al., and Shin et al. discovered that repeat head CT are advantageous in a subset of patients with a higher risk of ICH progression, even in the absence of neurological deterioration [14,29,30,31]. Cepeda et al. concluded that older age, the initial volume of the hemorrhage, SDH, and the mechanism of trauma are all associated with an increased risk of ICH progression [14]. Therefore, in this study, a repeat head CT was prescribed to patients with a higher risk of ICH progression; patients whose ages are >65 years old, taking long-term antiplatelets or anticoagulants, or who had worrisome ICH (EDH/SAH/SDH > 5 mm) or multiple ICH.

A prospective cohort study was conducted in a level II trauma center, Hospital Tengku Ampuan Rahimah (HTAR), which has the second-highest number of trauma cases in Malaysia. We demonstrate that a repeat head CT has limited value in patients with no neurological deterioration despite having a higher risk of ICH progression. Additionally, we produce identical data outcomes as other studies. The intervention rate for MTBI patients with a higher risk of ICH progression and no neurological changes was 0.52%, comparable to the 0–1.5% reported in previous studies by Adatia et al. and others for all patients with MTBI [5,7,10,32,33]. Furthermore, 17.9% of patients had ICH progression on the scheduled repeat head CT. We found no statistical significance when comparing the surgical intervention rate for those with ICH progression in scheduled repeat head CT to those without. In patients without neurological deterioration, the repeat head CT appears to have no diagnostic value when performed routinely [34].

In the study, only one patient on long-term aspirin required surgical intervention among the 34 patients with ICH progression. The 72-year-old patient showed no neurological deterioration, but the repeat head CT showed enlargement of the existing SDH with midline shift. Numerous studies [19,20,29,35,36,37,38] have found that patients taking anticoagulants and antiplatelets are more likely to develop delayed ICH. Nonetheless, our sub-analysis revealed no association between antiplatelet use and progression of ICH. This result was consistent with an observational study by Bauma et al. [38] in which only 1% of elderly patients experienced delayed ICH without clinically significant neurological deterioration. The study was conducted on a population in which the initial head CT revealed no ICH.

This study also achieves both secondary outcomes. Our data demonstrate that patients with ICH progression did not have an increased risk of neurological deterioration during admission compared to those without ICH progression (5.9% versus 5.2%, respectively). Simultaneously, patients with ICH progression also did not have a higher rate of readmission than those without (*p* = 0.97, 5.9% versus 3.9%).

The advantage of repeating a scan must outweigh the risk. Anandalwar et al. and Sifri et al. agreed repeating head CT in patients with no neurological deterioration may not be necessary if there is no change in management or outcome [6,13]. While Brenner et al. and Kumar et al. confirmed that head CT is useful for evaluating and monitoring ICH progression, there are risks associated with repeated exposure, including radiation exposure and increased healthcare costs [39,40,41].

Gerard et al. reported that small ICHs tend to expand after a few hours due to the absence of a tamponade effect to overcome the bleeding vessel pressure [42]. MTBI patients typically stop bleeding after 36 h, i.e., when hemostasis is achieved through thrombosis [42]. Therefore, even if ICH progression occurs initially, the need for surgical intervention in MTBI is minimal. In contrast, in cases of moderate to severe TBI, especially polytrauma, the ICH expands until it achieves the tamponade effect due to the dysfunctional coagulation cascade caused by clotting factor consumption [43]. This consistently results in significant mass effect and increased intracranial pressure (ICP). Hence, surgical intervention is required to stop the bleeding and reduce the ICP.

Several studies indicate that contusional bleeding is more likely to progress [18,27,44]. The pathophysiology of contusion progression is due to the metabolically compromised ‘traumatic penumbra’ surrounding the contusion core, making it susceptible to secondary insults [32]. As a result of bone impact, contusions associated with TBI are commonly observed in the frontal and temporal lobes but may also be found in other regions of the brain [18]. Adatia et al. discovered a few circumstances in which contusions were more likely to progress; frontal contusions are 1.5 times more likely to progress, contrecoup contusions are twice as likely, and patients with bilateral or multiple contusions are three times more likely to progress. In contrast, our study found no correlation between ICH subtypes and the progression of ICH in patients with mild TBI. Instead of the type of ICH, the initial GCS was found to be a predictor of ICH progression [31].

Consequentially, we observed patients with ICH progression until day five post-trauma and performed a repeat head CT prior to discharge to ensure the existing bleed had no further expansion. The repeated head CT five days after trauma typically showed improvement or resolution resulting from fibrinolysis, which decreased hematoma density. Our sub-analysis revealed that the length of stay for patients with ICH progression was significantly longer (5.7 days versus 4.2 days, *p* = 0.001), owing to the need for repeat head CT post-trauma day five. It was determined that prolonged hospitalization and repeat head CT brain on day five post-trauma were unnecessary, given that there was no change in management.

We discovered that, on average, 4.2% of MTBI patients were readmitted within two weeks of their initial discharge. Their readmission was primarily attributable to persistent symptoms, such as headache and dizziness. Repeated head CT on these patients revealed no ICH progression. Consequently, no significant correlation was found between the readmission rate and ICH progression, indicating that the duration of symptoms is unrelated to ICH progression [45,46,47]. Headache was the leading cause of readmissions, but the pathophysiology of post-traumatic headache is not well understood [34,48,49]. Firas et al. [50] reported that these headaches were associated with axonal injury and subsequent neurophysiological changes resulting from the inertial force of traumatic brain injury. As the disturbances occur at the neuronal and synaptic levels, head CT has a limited role in detecting these subtle injuries [45]. Compared to T2-weight MRI, which has a sensitivity of 92% for detecting axonal injury, the CT brain has only a 19% sensitivity [51].

In this study, two young patients exhibited neurological deterioration and decline in GCS prior to the scheduled repeat head CT. Both had their initial head CT performed within 6 h post-trauma. Even though our study found no association between the timing of the first head CT and the progression of ICH, Sifri et al., Narayan et al., Collins et al., and Yadav et al. recommended that a repeat head CT should be performed on TBI patients whose initial scan was performed within 6 h of the trauma, particularly those with EDH [10,21,23,24]. ICH requires hours to accumulate and form clots after vessel tearing; hence it may not be detected on the initial head CT [34]. Washington et al., Sifri et al., Wang et al., and Kaups et al. demonstrated no clear indication or recommendation for repeating a head CT in MTBI patients [7,10,12,52]. Moreover, the current research focuses primarily on patients with moderate to severe TBI who have difficulty monitoring their GCS [11].

## 5. Limitations

There are several limitations in this single-center, prospective, non-randomized observational study. The sample size is small and collected from a single institution. Second, further long-term follow-up should monitor the patient’s outcome. Furthermore, this study assumes that qualified and vigilant staff are available to perform frequent and reliable neurologic examinations at a Level II trauma center. If adequate neurologic monitoring is not available, the recommendations of this study are not applicable.

Each subgroup, such as elderly patients, patients on antiplatelet or anticoagulant therapy, patients with EDH or SDH, or multiple ICH, should be studied separately for better analysis and interpretation. To date, there is no clearly defined literature on the non-surgical management of elderly patients on antiplatelet or anticoagulant therapy with ICH. Further research into the progression of this subgroup would aid in determining the management strategy for the interval and frequency of repeating CT brain.

We limited our outcome measures to the intervention and readmission rate due to time constraints and small sample size. To weigh the benefit of scheduled repeat CT brain, we propose a study on other outcomes, such as the patient’s functional status, cost, and radiation exposure. Multicenter prospective studies are needed to further reinforce and validate these conclusions and reduced unnecessary head CT scans.

## 6. Conclusions

In this study, ICH progression detected in scheduled repeat head CT in patients with mild TBI has no impact on the patient’s clinical outcome or management. There is no difference in intervention and readmission rates between MTBI patients with and without ICH progression. These results suggest that repeat head CT in MTBI patients with no neurological deterioration is not recommended, even in those with a higher risk of ICH progression.

More research on each subgroup of patients is necessary to determine the benefit of a scheduled repeat head CT, as some neurosurgeons may be uneasy with this practice due to the possibility of missing an undetected worsening brain injury that may require neurosurgical intervention.

Decreasing the number of additional CT scans could result in reduced hospital costs, decreased staff and patient exposure to radiation, and fewer problems with patient transportation and total length of stay without compromising the safety of patients.

## Figures and Tables

**Figure 1 ijerph-19-14311-f001:**
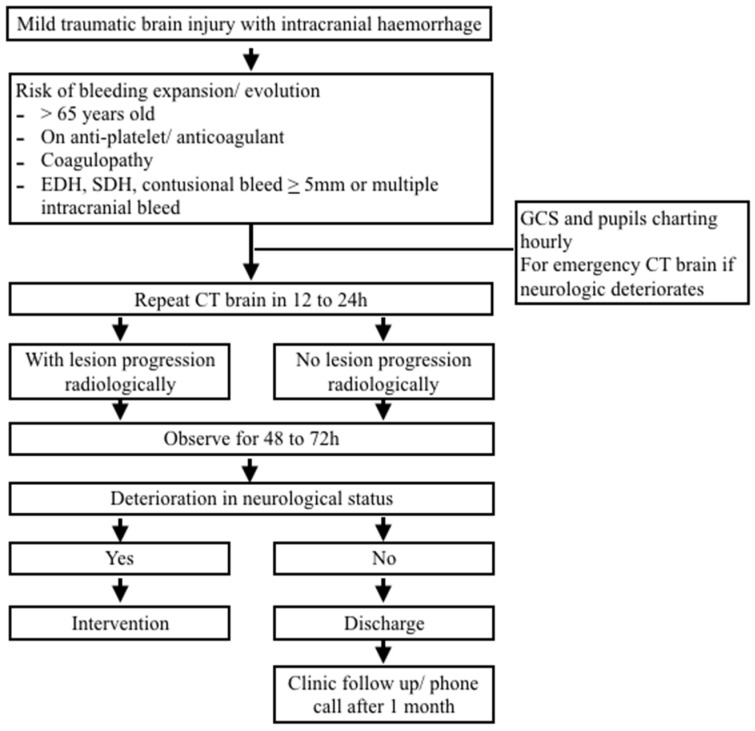
Study flow diagram.

**Table 1 ijerph-19-14311-t001:** Demographic characteristics of the patients in relation to ICH progression.

Demographic Characteristic	Number of Patients with ICH Progression, *n* (%)	Number of Patients without ICH Progression, *n* (%)
**Gender**		
Male	24 (70.6)	113 (72.9)
Female	10 (29.4)	42 (27.1)
**Age**		
<65 years old	23 (67.6)	110 (70.9)
≥65 years old	11 (32.4)	45 (29.1)
**Use of antiplatelet**		
On antiplatelet	6 (17.6)	19 (12.3)
Not on antiplatelet	28 (82.4)	136 (87.7)
**Mechanism of injury**		
Motor-vehicle accident	18 (52.9)	114 (73.5)
Fall	11 (32.4)	32 (20.7)
Other	5 (14.7)	9 (5.8)
**Total**	**34 (100.0)**	**155 (100.0)**

**Table 2 ijerph-19-14311-t002:** Baseline ICH in the first CT in relation with ICH progression (F = 0.25, *p* = 0.62).

Findings of First CT Brain	Number of Patients with ICH Progression, *n* (%)	Number of Patients with No ICH Progression, *n* (%)
Number of SDH, *n* (%)	10 (29.5)	49 (32.6)
Number of EDH, *n* (%)	1 (2.9)	6 (3.9)
Number of contusional bleed, *n* (%)	3 (8.8)	27 (17.4)
Number of SAH, *n* (%)	1 (2.9)	10 (6.5)
Number of multiple ICH, *n* (%)	19 (55.9)	63 (40.6)
**Total number of patients, *n* (%)**	**34 (100.00)**	**155 (100.0)**

**Table 3 ijerph-19-14311-t003:** Progression of ICH in scheduled repeat CT in relation to intervention.

Progression of Bleed in Scheduled Repeat CT	Intervention		
	Number with Intervention, *n* (%)	Number with No Intervention, *n* (%)	Total (%)
Number with ICH progression, *n* (%)	1 (2.9)	33 (97.1)	34 (100)
Number with no ICH progression, *n* (%)	0 (0.0)	155 (100.0)	155 (100)
**Total, *n* (%)**	1 (0.5)	188 (99.5)	189 (100)

**Table 4 ijerph-19-14311-t004:** Timing of the first CT brain in relation to ICH progression in scheduled repeat CT brain.

Timing of the First CT	Progression of ICH in Scheduled Repeat CT		
	Number with ICH Progression, *n* (%)	Number with No ICH Progression, *n* (%)	Total Number, *n* (%)
Number of first CT brain done within 6 h, *n* (%)	15 (20.5)	58 (79.5)	73 (100.00)
Number of first CT brain done after 6 h, *n* (%)	22 (18.5)	97 (81.5)	119 (100.0)
**Total, *n* (%)**	37 (19.3)	155 (80.7)	192 (100.0)

**Table 5 ijerph-19-14311-t005:** Radiological progression of ICH and its relationship with neurological deterioration during admission.

Progression of Bleed in Scheduled Repeat CT	Neurological Deterioration after Scheduled Repeat CT		
	Number with Neurological Deterioration, *n* (%)	Number with No Neurological Deterioration, *n* (%)	Total (%)
Number with ICH progression, *n* (%)	2 (5.9)	32 (94.1)	34 (100)
Number with no ICH progression, *n* (%)	8 (5.2)	147 (94.2)	155 (100)
**Total, *n* (%)**	10 (5.3)	179 (94.7)	189 (100)

**Table 6 ijerph-19-14311-t006:** Progression of ICH in scheduled repeat CT brain in relation to readmission of the patients.

Progression of ICH in Scheduled Repeat CT	Readmission		
	Number of Readmission, *n* (%)	Number with No Readmission, *n* (%)	Total (%)
Number with ICH progression, *n* (%)	2 (5.9)	32 (94.1)	34 (100)
Number with no ICH progression, *n* (%)	6 (3.9)	147 (96.1)	153 (100)
**Total, *n* (%)**	8 (4.2)	181 (95.8)	189 (100)

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
