# Peer review of "Value of Repeat CT Brain in Mild Traumatic Brain Injury Patients with High Risk of Intracerebral Hemorrhage Progression"

_ijerph, 2022, doi:10.3390/ijerph192114311_

Round 1

Reviewer 1 Report

This paper presents data collected on repeat CT images for adults with mild TBI.

There was some lack of clarity in the paper with respect to the aim of the paper (which not only changes from abstract, method and discussion, but it is also not clear if it was determined apriori).

The method was unclear - were patients opportunistically included, or consecutive cases? Were only those who deteriorated given second CTs? Who are the admitted mTBI cohort in Malaysia (this differs across health systems)? 

I suggest that the paper should adhere to the Strengthening the Reporting of Observational Studies in Epidemiology (STROBE) Statement: guidelines for reporting observational studies. This will ensure that the paper clearly articulates method. In addition, the abstract (and introduction) should be tightened to focus on the importance of considering anticoagulation in older patients who present with mTBI; for example line 90-91. Is this the cohort? And then a subcohort is the older patients (lines 92-94)? Or only the older patients? Using STROBE will assist to gain clarity in the observational cohort.

There are also some important TBI distinctions that should be made in your paper. GCS 13-15 is mild, but to classify as deteriorating they have to drop 2 points- thus, the cohort would no longer be considered mild. 

Finally the discussion needs greater clarity back to the age and cohort. For example, if intervention is based on symptoms (as implied in paragraph 2 of the discussion), then you should outline the criteria. This should feature in the method (not being raised in the discussion). I also believe that the progression of a bleed is the most important discussion point, so only having a small component of the discussion on this (paragraph 4) was disappointing. I believe the clinical implications of this work could be made clearer to the reader.

Overall this could be an interesting paper for clinicians who manage older adults who suffer a mTBI and are on anticoag meds. However I believe that the paper does need to be restructured to reflect the STROBE guidelines and present information appropriately for readers.

Reviewer 2 Report

ijerph-1933408 „Value of Repeat CT Brain in Mild Traumatic Brain Injury Patients with High Risk of Intracranial Hemorrhage Progression“

The authors present a prospective single center study including 192 mild TBI patients, who received a scheduled repeat CT. The rate of bleeding progression was recorded (18%). The intervention and readmission rate of patients with and without radiological hematoma progression did not differ significantly. The authors conclude that scheduled repeat CT for patients without neurological deterioration is not required.

The authors present a well conducted and very valuable study in order to reduce the routine use of head CT, which might be especially interesting in areas without an easy access to CT.

Specific Comments:

·      General: abbreviations need to be introduced again in the main manuscript; number of decimal places should be reduced, some language editing (l37 CT brain replaced by cranial CT).

·      There exist a variety of definitions for mild TBI in the literature, especially which GCS limits should be used. While the authors use GCS 13-15 other authors use GCS 14 and 15 (Stein and Spettell 1995, Ingebrigtsen, Romner et al. 2000). Especially for the purpose of the presented study, a subgroup analysis for these different cohorts depending on GCS might be interesting.

·      The abbreviation ICH is normally used for intracerebral hemorrhage. However, the authors use it for intracranialhemorrhage. This might cause some confusion especially for a reader not being familiar with TBI/ Neurotrauma terminology/topics. The authors should consider to abstain from using ICH.

·      Please add who assessed the GCS and neurological status with respect to the physician’s speciality as well as rank (student, resident, consultant). Both are highly subjective and examiner dependent (Reith, Van den Brande et al. 2016).

·      Was the CT analysed by a radiologist or neurosurgeon?

·      Please check english spelling. E.g.(l173): „patients with ICH progression did not have a higher readmission rate than those who did not“, instead: „than those without“ 

Ingebrigtsen, T., B. Romner and C. Kock-Jensen (2000). "Scandinavian guidelines for initial management of minimal, mild, and moderate head injuries. The Scandinavian Neurotrauma Committee." J Trauma 48(4): 760-766.

Reith, F. C., R. Van den Brande, A. Synnot, R. Gruen and A. I. Maas (2016). "The reliability of the Glasgow Coma Scale: a systematic review." Intensive Care Med 42(1): 3-15.

Stein, S. C. and C. Spettell (1995). "The Head Injury Severity Scale (HISS): a practical classification of closed-head injury." Brain Inj 9(5): 437-444.

Reviewer 3 Report

Thank you for the opportunity to review the article titled: Value of Repeat CT Brain in Mild Traumatic Brain Injury Patients with High Risk of Intracranial Hemorrhage Progression.

The article is interesting but demands some corrections.

Abstract:

It seems a little bit chaotic. Please, consider reordering. First, what, and why investigators are checking? Second, what is the studied group, and which are the methods of investigation? Dates are not mandatory; the abstract may be much more straight to the point. Further, the most important results and conclusions. Please, enhance the benefits of the research.

Introduction:

The Introduction is informative, but some worldwide available data are needed. The Authors focused their attention on their country, while the worldwide statistics are also important. Comparisons should be placed both in the Introduction and Discussion sections. It seems to be highly important to avoid limiting the report to one region only. Moreover, a little bit more data regarding the disease itself could be valuable and helpful in building the logic flow.

The Introduction section should provide the Reader with the Author's “train of thoughts” – what and why we are investigating, why we believe it is serious and worth investigating, what are our objectives (main and more detailed), etc.

I believe that references in the brackets are part of the sentence, so the phrase completes after, not before the references are provided.

The Introduction section mixes Introduction, Methods, and Discussion. Please, discuss the other results in the Discussion. The Introduction overviews the research. Please, transfer data from other studies (i.e. 179 patients…) to the Discussion.

Finally, the Introduction is too long…it should be substantially shortened, and most of the content should be placed in the Discussion section. Focus on the background, do not discuss nor present Methods, Results, and conclusions with the theoretical overview.

Materials and Methods:

The first sentence is more appropriate for the Discussion section. It explains the results probably. Or justifies the conclusions. But it does not benefit the Methods section.

Please, consider changing the study design to the original article instead of prospective. The study description does not reflect the longitudinal, cohort study design.

Please, indicate CLEARLY and in one, consistent paragraph all the inclusion/exclusion criteria, studies protocol, types of lesions, monitored disease, investigators, etc. Please, make sure you indicated clearly how you conducted the research in detail and how you would like to move from there with your study.

Statistical analysis is more complex than the Authors described. Please, make sure you explained it step-by-step and mentioned all the most relevant observations in the Results section. Please, make sure you indicated pathway and software in Methods, and values in Results.  Please, describe all tests you have used to evaluate your database.

Ethics should open the Methods section.

Results:

Please, make sure that the material is well-presented in the Materials and Methods, not in the Results section. Moreover, the Results suggest that the study is retrospective, not prospective. Please, consider changing the format of your study description. Please, make sure you presented epidemiological data in Methods, not in Results.

Results section demands very detailed and substantial reorganization of the content and substantial recorrection. Please, describe your findings, and discuss those in the Discussion section.

Discussion:

The section deserves attention. It should be adjusted with the above-mentioned data. Discussion should justify or reject the study findings; the study limitations and perspectives should be also mentioned. Please, carefully evaluate the content, exclude elements which are not applicable to the Discussion, mention objectives, results, and comparisons and provide valid conclusions.

Round 2

Reviewer 3 Report

Authors provided several corrections, the article has been improved.